# Exact Rate-Distortion in Autoencoders via Echo Noise

**Rob Brekelmans, Daniel Moyer, Aram Galstyan, Greg Ver Steeg**
Information Sciences Institute
University of Southern California
Marina del Rey, CA 90292
`brekelma, moyerd@usc.edu; galstyan, gregv@isi.edu`

## Abstract

Compression is at the heart of effective representation learning. However, lossy compression is typically achieved through simple parametric models like Gaussian noise to preserve analytic tractability, and the limitations this imposes on learning are largely unexplored. Further, the Gaussian prior assumptions in models such as variational autoencoders (VAEs) provide only an upper bound on the compression rate in general. We introduce a new noise channel, *Echo noise*, that admits a simple, exact expression for mutual information for arbitrary input distributions. The noise is constructed in a data-driven fashion that does not require restrictive distributional assumptions. With its complex encoding mechanism and exact rate regularization, Echo leads to improved bounds on log-likelihood and dominates $\beta$-VAEs across the achievable range of rate-distortion trade-offs. Further, we show that Echo noise can outperform flow-based methods without the need to train additional distributional transformations.

## 1   Introduction

Rate-distortion theory provides an organizing principle for representation learning that is enshrined in machine learning as the Information Bottleneck principle [39]. The goal is to compress input random variables $X$ into a representation $Z$ with mutual information rate $I(X; Z)$, while minimizing a distortion measure that captures our ability to use the representation for a task. For the rate to be restricted, some information must be lost through noise. Despite the use of increasingly complex encoding functions via neural networks, simple noise models like Gaussians still dominate the literature because of their analytic tractability. Unfortunately, the effect of these assumptions on the quality of learned representations is not well understood.

The Variational Autoencoding (VAE) framework [21, 36] has provided the basis for a number of recent developments in representation learning [1, 10, 11, 18, 20, 41]. While VAEs were originally motivated as performing posterior inference under a generative model, several recent works have viewed the Evidence Lower Bound objective as corresponding to an unsupervised rate-distortion problem [1, 3, 35]. From this perspective, reconstruction of the input provides the distortion measure, while the KL divergence between encoder and prior gives an upper bound on the information rate that depends heavily on the choice of prior [3, 37, 40].

In this work, we deconstruct this interpretation of VAEs and their extensions. Do the restrictive assumptions of the Gaussian noise model limit the quality of VAE representations? Does forcing the latent space to be independent and Gaussian constrain the expressivity of our models? We find evidence to support both claims, showing that a powerful noise model can achieve more efficient lossy compression and that relaxing prior or marginal assumptions can lead to better bounds on both the information rate and log-likelihood.

The main contribution of this paper is the introduction of the Echo noise channel, a powerful, data-driven improvement over Gaussian channels whose compression rate can be precisely expressed for

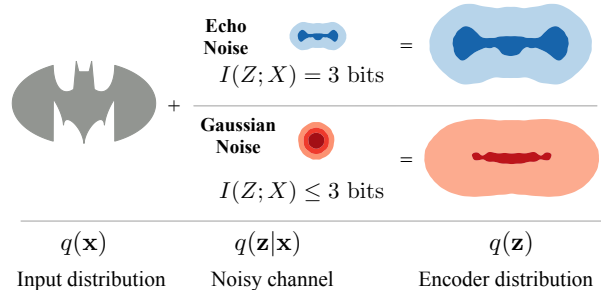

Figure 1: For a noisy channel characterized by $\mathbf{z} = \mathbf{x} + s\epsilon$, we compare drawing the noise, $\epsilon$, from a Gaussian distribution (as in VAEs) or an Echo distribution.

arbitrary input distributions. Echo noise is constructed from the empirical distribution of its inputs, allowing its variation to reflect that of the source (see Fig. 1). We leverage this relationship to derive an analytic form for mutual information that avoids distributional assumptions on either the noise or the encoding marginal. Further, the Echo channel avoids the need to specify a prior, and instead implicitly uses the optimal prior in the Evidence Lower Bound. This marginal distribution is neither Gaussian nor independent in general.

After introducing the Echo noise channel and an exact characterization of its information rate in Sec. 2, we proceed to interpret Variational Autoencoders from an encoding perspective in Sec. 3. We formally define our rate-distortion objective in Sec. 3.1, and draw connections with recent related works in Sec. 4. Finally, we report log likelihood results, visualize the space of compression-reconstruction trade-offs, and evaluate disentanglement in Echo representations in Sec. 5.

## 2 Echo Noise

To avoid learning representations that memorize the data, we would like to constrain the mutual information between the input $X$ and the representation $Z$. Since we have freedom to choose how to encode the data, we can design a noise model that facilitates calculating this generally intractable quantity.

The Echo noise channel has a shift-and-scale form that mirrors the reparameterization trick in VAEs. Referring to the observed data distribution as $q(\mathbf{x})$, with $\mathbf{z} \in \mathbb{R}^{d_z}, \mathbf{x} \in \mathbb{R}^{d_x}$, we can define the stochastic encoder $q_\phi(\mathbf{z}|\mathbf{x})$ using:

$$\mathbf{z} = f(\mathbf{x}) + S(\mathbf{x})\epsilon \tag{1}$$

For brevity, we omit the subscripts that indicate that the functions $f : \mathbb{R}^{d_x} \to \mathbb{R}^{d_z}$ and matrix function $S : \mathbb{R}^{d_x} \to \mathbb{R}^{d_z} \times \mathbb{R}^{d_z}$ depend on neural networks parameterized by $\phi$. All that remains to specify the encoder is to fix the distribution of the noise variable, $q(\epsilon)$. For VAEs, the noise is typically chosen to be Gaussian, $\epsilon \sim \mathcal{N}(0, \mathbb{I}_{d_z})$. [1]

With the goal of calculating mutual information, we will need to compare the marginal entropy $H(Z)$, which integrates over samples $\mathbf{x}$, and the conditional entropy $H(Z|X)$, whose stochasticity is only due to the noise for deterministic $f(\mathbf{x})$ and $S(\mathbf{x})$. The choice of noise will affect both quantities, and our approach is to relate them by enforcing an equivalence between the distributions $q(\mathbf{z})$ and $q(\epsilon)$.

Since $q(\mathbf{z}) = \int q_\phi(\mathbf{z}|\mathbf{x})q(\mathbf{x})d\mathbf{x}$ is defined in terms of the source, we can also imagine constructing the noise in a data-driven way. For instance, we could draw $\epsilon = f(x'), x' \overset{iid}{\sim} q(\mathbf{x})$ in an effort to make the noise match the channel output. However, this changes the distribution of $Z$ and the noise would need to be updated to continue resembling the output.

Instead, by iteratively applying Eq. 1, we can guarantee that the noise and marginal distributions match in the limit. Using superscripts to indicate iid samples $\mathbf{x}^\ell \overset{iid}{\sim} q(\mathbf{x})$, we draw $\epsilon$ according to:

$$
\begin{aligned}
\epsilon &= f(\mathbf{x}^0) + S(\mathbf{x}^0)\Big( f(\mathbf{x}^1) + S(\mathbf{x}^1)\Big( f(\mathbf{x}^2) + S(\mathbf{x}^2)(... \\
&= f(\mathbf{x}^0) + S(\mathbf{x}^0)f(\mathbf{x}^1) + S(\mathbf{x}^0)S(\mathbf{x}^1)f(\mathbf{x}^2)...
\end{aligned}
\tag{2}
$$

Echo noise is thus constructed using an infinite sum over attenuated "echoes" of the transformed data samples. This can be written more compactly as follows.

**Definition: Echo Noise** The Echo noise distribution $E(f(\mathbf{x}), S(\mathbf{x}), q(\mathbf{x}))$ is defined for functions $f, S$, and probability density function $q$ over $\mathbf{x} \in \mathbb{R}^{d_x}$, by sampling according to the following procedure.

$$
\epsilon = \sum_{\ell=0}^\infty \left( \prod_{\ell'=1}^\ell S(\mathbf{x}^{\ell'}) \right) f(\mathbf{x}^\ell), \qquad \mathbf{x}^\ell \overset{iid}{\sim} q(\mathbf{x})
\tag{3}
$$

Although the noise distribution may be complex, it has the interesting property that it exactly matches the eventual output marginal $q_\phi(\mathbf{z})$.

**Lemma 2.1** (Echo noise matches channel output). *If $\epsilon \sim Echo(f(\mathbf{x}), S(\mathbf{x}), q(\mathbf{x}))$ and $\mathbf{z} = f(\mathbf{x}) + S(\mathbf{x})\epsilon$, then $\mathbf{z}$ has the same distribution as $\epsilon$.*

We can observe this relationship by simply re-labeling the sample indices in the expanded expression for the noise in Eq. 2. In particular, the training example that we condition on in Eq. 1 corresponds to the first sample $\mathbf{x}^0$ in a draw from the noise. This equivalence is the key insight leading to an exact expression for the mutual information:

**Theorem 2.2** (Echo Information). *For any source distribution $q(\mathbf{x})$, and a noisy channel defined by Eq. 1 that satisfies 2.3, the mutual information is as follows:*

$$
I(X; Z) = -\mathbb{E}_{\mathbf{x}} \log |\det S(\mathbf{x})|
\tag{4}
$$

*Proof.* We start by expanding the definition of mutual information in terms of entropies. Since $f(\mathbf{x})$ and $S(\mathbf{x})$ are deterministic, we treat them as constants after conditioning on $X = \mathbf{x}$. The stochasticity underlying $H(Z|X = \mathbf{x})$ is thus only due to the random variable $\epsilon$.

$$
\begin{aligned}
I(X; Z) &= H(Z) - H(Z|X) \\
&= H(Z) - \mathbb{E}_{\mathbf{x}} H(f(\mathbf{x}) + S(\mathbf{x})\mathcal{E} \mid X = \mathbf{x}) \\
&= H(Z) - \mathbb{E}_{\mathbf{x}} H(S(\mathbf{x})\mathcal{E} \mid X = \mathbf{x}) \\
&= H(Z) - H(\mathcal{E}) - \mathbb{E}_{\mathbf{x}} \log |\det S(\mathbf{x})| \\
&= -\mathbb{E}_{\mathbf{x}} \log |\det S(X)|
\end{aligned}
$$

We have used the translation invariance of differential entropy in the third line, and the scaling property in the fourth line [12]. The entropy terms cancel as a result of Lemma 2.1. $\square$

In this work, we consider only diagonal $S(\mathbf{x}) \equiv \mathrm{diag}(s_1(\mathbf{x}), \dots, s_{d_z}(\mathbf{x}))$ as is typical for VAEs, so that the determinant in Eq. 4 simplifies as $I(X; Z) = -\sum_j \mathbb{E}_{\mathbf{x}} \log |s_j(\mathbf{x})| = \sum_j I(X; Z_j)$.

Finally, we note that the noise distribution $q(\epsilon)$ is only defined implicitly through a sampling procedure. For this to be meaningful, we must ensure that the infinite sum converges.

**Lemma 2.3.** *The infinite sum in Eq. 3 converges, and thus Echo noise sampling is well-behaved, if $\forall \mathbf{x}, \exists M$ s.t. $|f(\mathbf{x})| \leq M$ and $\rho(S(\mathbf{x})) < 1$, where $\rho$ is the spectral radius.*

In App. B, we discuss several implementation choices to guarantee that these conditions are met and that Echo noise can be accurately sampled using a finite number of terms. This is particularly difficult in the high noise, low information regime, as zero mutual information $(s_j(\mathbf{x})) = 1 \forall \mathbf{x}, j)$ would imply an infinite amount of noise. To avoid this issue and ensure precise sampling, we clip the magnitude of $s_j(\mathbf{x})$ so that, for a given $M$ and number of samples, the sum of remainder terms is guaranteed to be within machine precision. This imposes a lower bound on the achievable rate across the Echo channel, which depends on the number of terms considered and can be tuned by the practitioner.

## 2.1 Properties of Echo Noise

We can visualize applying Echo noise to a complex input distribution in Fig. 1, using the identity transformation $f(\mathbf{x}) = \mathbf{x}$ and constant noise scaling $s_j(\mathbf{x}) = .5$. Here, we directly observe the equivalence of the noise and output distributions. Further, the data-driven nature of the Echo channel means it can leverage the structure in the (transformed) input to destroy information in a more targeted way than spherical Gaussian noise.

In particular, Echo's ability to add noise that is correlated across dimensions distinguishes it from common diagonal noise models. It is important to note that the noise still reflects the dependence in $f(\mathbf{x})$ even when $S(\mathbf{x})$ is diagonal. In fact, we show in App. C that $TC(Z) = TC(Z|X)$ for the diagonal case, where total correlation measures the divergence from independence, e.g. $TC(Z|X) = D_{KL}[q(\mathbf{z}|\mathbf{x})||\prod q(z_j|\mathbf{x})]$ [43].

In the setting of learned $f(\mathbf{x})$ and $S(\mathbf{x})$, notice that the noise depends on the parameters. This means that training gradients are propagated through $\epsilon$, unlike traditional VAEs where $q(\epsilon)$ is fixed. This may be a factor in improved performance: data samples are used as both signal and noise in different parts of the optimization, leading to a more efficient use of data.

Finally, the Echo channel fulfills several of the desirable properties that often motivate Gaussian noise and prior assumptions. Eqs. 1 and 3 define a simple sampling procedure that only requires a supply of iid samples from the input distribution. It is easy to sample both the noise and conditional distributions for the purposes of evaluating expectations, while Echo also provides a natural way to sample from the true encoding marginal $q_\phi(\mathbf{z})$ via its equivalence with $q(\epsilon)$. While we cannot evaluate the density of a given $\mathbf{z}$ under $q_\phi(\mathbf{z}|\mathbf{x})$ or $q_\phi(\mathbf{z})$, as might be useful in importance sampling [8], we can characterize their relationship *on average* using the mutual information in Eq. 4. These ingredients make Echo noise useful for learning representations within the autoencoder framework.

## 3 Lossy Compression in VAEs

Variational Autoencoders (VAEs) [21, 36] seek to maximize the log-likelihood of data under a latent factor generative model defined by $p_\theta(\mathbf{x}, \mathbf{z}) = p(\mathbf{z})p_\theta(\mathbf{x}|\mathbf{z})$, where $\theta$ represents parameters of the generative model decoder and $p(\mathbf{z})$ is the prior distribution over latent variables. However, maximum likelihood is intractable in general due to the difficult integral over $Z$, $\log p_\theta(\mathbf{x}) = \log \int p(\mathbf{z})p_\theta(\mathbf{x}|\mathbf{z})d\mathbf{z}$.

To avoid this problem, VAEs introduce a variational distribution, $q_\phi(\mathbf{z}|\mathbf{x})$, which encodes the input data $q(\mathbf{x})$ and approximates the generative model posterior $p_\theta(\mathbf{z}|\mathbf{x})$. This leads to the tractable (average) Evidence Lower Bound (ELBO) on likelihood:

$$\begin{aligned} \mathbb{E}_q \log p_\theta(\mathbf{x}) &\geq \mathbb{E}_q \log p_\theta(\mathbf{x}) - D_{KL}[q_\phi(\mathbf{z}|\mathbf{x})||p_\theta(\mathbf{z}|\mathbf{x})] \\ &= \mathbb{E}_q \log p_\theta(\mathbf{x}|\mathbf{z}) - D_{KL}[q_\phi(\mathbf{z}|\mathbf{x})||p(\mathbf{z})] \end{aligned} \tag{5}$$

The connection between VAEs and rate-distortion theory can be seen using a decomposition of the KL divergence term from Hoffman and Johnson [19].

$$\begin{aligned} D_{KL}[q_\phi(\mathbf{z}|\mathbf{x})||p(\mathbf{z})] &= D_{KL}[q_\phi(\mathbf{z}|\mathbf{x})||q_\phi(\mathbf{z})] + D_{KL}[q_\phi(\mathbf{z})||p(\mathbf{z})] \\ &\geq D_{KL}[q_\phi(\mathbf{z}|\mathbf{x})||q_\phi(\mathbf{z}))] = I_q(X; Z) \end{aligned} \tag{6}$$

This decomposition lends insight into the orthogonal goals of the ELBO regularization term. The mutual information $I_q(X; Z)$ encourages lossy compression of the data into a latent code, while the marginal divergence enforces consistency with the prior. The non-negativity of the KL divergence implies that each of these terms detracts from our likelihood bound.

Similarly, we observe that $D_{KL}[q_\phi(\mathbf{z}|\mathbf{x})||p(\mathbf{z})]$ gives an upper bound on the mutual information, with a gap of $D_{KL}[q_\phi(\mathbf{z})||p(\mathbf{z})]$. From this perspective, a static Gaussian prior can be seen a particular and possibly loose marginal approximation [3, 14, 37]. The true encoding marginal $q_\phi(\mathbf{z})$ provides the unique, optimal choice of prior and leads to a tighter bound on the likelihood:

$$\mathbb{E}_q \log p_\theta(\mathbf{x}) \geq \mathbb{E}_q \log p_\theta(\mathbf{x}|\mathbf{z}) - I_q(X; Z) \tag{7}$$

Our exact expression for the mutual information over an Echo channel provides the first general method to directly optimize this objective. This corresponds to adaptively setting $p(\mathbf{z})$ equal to $q_\phi(\mathbf{z})$

throughout training, so that Eq. 7 can be seen as bounding the likelihood under the generative model $p(\mathbf{x}) = \int q_\phi(\mathbf{z}) p_\theta(\mathbf{x}|\mathbf{z}) d\mathbf{x}$.

## 3.1 Rate-Distortion Objective

While the VAE is motivated as performing amortized inference of the latent variables in a generative model, the prior is rarely leveraged to encode domain-specific structure. Further, we have shown that enforcing prior consistency can detract from likelihood bounds.

We instead follow Alemi et al. [3] in advocating that representation learning be motivated from an encoding perspective using rate-distortion theory. In particular, we choose reconstruction under the generative model as the distortion measure $d(\mathbf{x}, \mathbf{z}) = -\log p_\theta(\mathbf{x}|\mathbf{z})$, and study the following optimization problem:

$$\max_{\theta, \phi} \mathbb{E}_{q_\phi} \log p_\theta(\mathbf{x}|\mathbf{z}) - \beta I_q(X; Z) \tag{8}$$

While this resembles the $\beta$-VAE objective of Higgins et al. [18], we highlight two notable distinctions. First, treating $I_q(X; Z)$ rather than the upper bound $D_{KL}[q_\phi(\mathbf{z}|\mathbf{x})||p(\mathbf{z})]$ avoids the need to specify a prior and facilitates a direct interpretation in terms of lossy compression. Further, the $\beta$ parameter is naturally interpreted as a Lagrange multiplier enforcing a constraint on $I_q(X; Z)$. The special choice of $\beta = 1$ gives a bound on log-likelihood according to Eq. 7, which we use to compare results across methods in Sec. 5. We direct the reader to App. A for a more formal treatment of rate-distortion.

# 4 Related Work

**Rate-Distortion Theory:** A number of recent works have made connections between the Evidence Lower Bound objective and rate-distortion theory [1, 3, 25, 35], with the average distortion corresponding to the cross entropy reconstruction loss as above.. In particular, Alemi et al. [3] consider the following upper and lower bounds on the mutual information $I_q(X; Z)$:

$$H - D = H_q(X) + \mathbb{E}_q \log p_\theta(\mathbf{x}|\mathbf{z}) \le I_q(X; Z) \le D_{KL}[q_\phi(\mathbf{z}|\mathbf{x})||r(\mathbf{z})] = R$$

With the data entropy as a constant, minimizing the cross entropy distortion corresponds to the variational information maximization lower bound of Barber and Agakov [4]. The upper bound matches the decomposition in Eq. 6 for the generalized choice of marginal $r(\mathbf{z})$. Several recent works have also considered 'learned priors' or flow-based density estimators [3, 11, 40] that seek to reduce the marginal divergence by approximating $q_\phi(\mathbf{z})$ (see below). Using this upper bound on the rate term, Alemi et al. [3] and Rezende and Viola [35] obtain objectives similar to Eq. 8.

Existing models are usually trained with a static $\beta$ [3, 18] or a heuristic annealing schedule [6, 9], which implicitly correspond to constant constraints (see App.A). However, setting target values for either the rate or distortion remains an interesting direction for future discussion. Rezende and Viola [35] view the distortion as an intuitive quantity to specify in practice, while Zhao et al. [46] train a separate model to provide constraint values. As both works show, specifying a constant and optimizing the Lagrange multiplier $\beta$ with gradient descent can lead to improved performance.

**Mutual Information in Unsupervised Learning:** A number of recent works have argued that the maximum likelihood objective may be insufficient to guarantee useful representations of data [3, 45]. In particular, when paired with powerful decoders that can match the data distribution, VAEs may learn to completely ignore the latent code [6, 11].

To rectify these issues, a commonly proposed solution has been to add terms to the objective function that maximize, minimize, or constrain the mutual information between data and representation [3, 7, 32, 45, 46]. However, justifications for these approaches have varied and numerous methods have been employed for estimating the mutual information. These include sampling [32], indirect optimization via other divergences [45], mixture entropy estimation [23], learned mixtures [40], autoregressive density estimation [3], and a dual form of the KL divergence [5]. Poole et al. [33] provide a thorough review and analysis of variational upper and lower bounds on mutual information, although recent results have shown limits on our ability to construct high confidence estimators directly from samples [29]. Echo notably avoids this limitation by providing an analytic expression for the rate whenever the representation is sampled according to Eq. 3.

Table 1: Test Log Likelihood Bounds

| Method | Binary MNIST | | | | Omniglot | | | | Fashion MNIST | | | | Params |
|---|---|---|---|---|---|---|---|---|---|---|---|---|---|
| | Rate | Dist | -ELBO | $\sigma$ | Rate | Dist | -ELBO | $\sigma$ | Rate | Dist | -ELBO | $\sigma$ | $(\cdot 10^6)$ |
| Echo | 26.4 | 62.4 | **88.8** | .18 | 30.2 | 84.4 | **114.6** | .30 | 16.6 | 218.3 | 234.9 | .21 | 1.40 |
| VAE | 26.2 | 63.6 | 89.8 | .18 | 30.5 | 86.5 | 117.0 | .44 | 15.7 | 219.3 | 235.0 | .10 | 1.40 |
| InfoVAE | 26.0 | 64.0 | 90.0 | .14 | 30.3 | 87.3 | 117.6 | .51 | 15.6 | 219.5 | 235.1 | .10 | 1.40 |
| VAE-MAF | 26.1 | 63.7 | 89.8 | .15 | 30.5 | 86.4 | 116.9 | .31 | 15.7 | 219.3 | 234.9 | .14 | 3.12 |
| VAE-Vamp | 26.3 | 63.0 | 89.3 | .19 | 30.8 | 84.3 | 115.1 | .28 | 15.9 | 218.5 | 234.4 | .08 | 1.99 |
| IAF-Prior | 26.5 | 63.5 | 90.0 | .13 | 30.5 | 86.7 | 117.2 | .36 | 15.8 | 219.1 | 234.9 | .10 | 3.12 |
| IAF+MMD | 26.3 | 63.6 | 90.1 | .15 | 30.7 | 86.4 | 117.1 | .28 | 15.7 | 219.2 | 234.9 | .13 | 3.12 |
| IAF-MAF | 26.4 | 63.6 | 89.9 | .18 | 30.6 | 86.5 | 117.1 | .24 | 15.8 | 219.1 | 234.9 | .14 | 4.84 |
| IAF-Vamp | 26.4 | 62.8 | 89.2 | .18 | 30.4 | 85.0 | 115.4 | .20 | 16.0 | 218.3 | **234.3** | .16 | 3.71 |

Among the approaches above,, the InfoVAE model of Zhao et al. [45] provides a potentially interesting comparison with our method. The objective adds a parameter $\lambda$ to more heavily regularize the marginal divergence and a parameter $\alpha$ to control mutual information. However, since $D_{KL}[q_\phi(\mathbf{z})||p(\mathbf{z})]$ is intractable, the Maximum Mean Discrepancy (MMD) [16] between the encoding outputs and a standard Gaussian is used as a proxy. For the choice of $\lambda = 1000$ (as in the original paper) and $\alpha = 0$ (no information preference), the objective simplifies to:

$$\mathcal{L}_{\text{InfoVAE}} = \mathcal{L}_{ELBO} - 999\, D_{\text{MMD}}[q_\phi(\mathbf{z})||p(\mathbf{z})]$$

The sizeable MMD penalty encourages $q_\phi(\mathbf{z}) \approx p(\mathbf{z})$, so that $D_{KL}[q_\phi(\mathbf{z}|\mathbf{x})||p(\mathbf{z})] \approx D_{KL}[q_\phi(\mathbf{z}|\mathbf{x})||q_\phi(\mathbf{z})] = I_q(X; Z)$. Thus, the KL divergence term in the ELBO should more closely reflect a mutual information regularizer, facilitating comparison with the rate in Echo models.

Flow models, which evaluate densities on simple distributions such as Gaussians but apply complex transformations with tractable Jacobians, are another prominent recent development in unsupervised learning [15, 22, 31, 34]. Flows can be used both as an encoding mechanism and marginal approximation for our purposes. In particular, Inverse Autoregressive Flow [22] can be seen as transforming the output of a Gaussian noise channel into an approximate posterior sample using a stack of autoregressive networks. Masked Autoregressive Flow [31] models a similar transformation with computational tradeoffs suited for density estimation, mapping latent samples to high probability under a Gaussian base distribution to approximate $q_\phi(\mathbf{z})$.

Finally, the VampPrior [40] may also be used as a marginal approximation, modeling $q_\phi(\mathbf{z})$ using a mixture distribution $\frac{1}{K} \sum_k q_\phi(\mathbf{z}|\mathbf{u}_k)$ evaluated on a set of 'pseudo-inputs' $\mathbf{u}_k \in \mathbb{R}^{d_x}$ learned by backpropagation.

# 5 Results

In this section, we would ideally like to quantify the impact of three key elements of the Echo approach: a data-driven noise model, exact rate regularization throughout training, and a flexible marginal distribution. In App. D.2, we observe that the dimension-wise marginals learned by Echo appear Gaussian despite our lack of explicit constraints. However, the *joint* marginal over $q_\phi(\mathbf{z})$ (or equivalently $q(\epsilon)$) may still have a complex dependence structure, which is not penalized for deviating from independence or Gaussianity. We calculate a second-order approximation of total correlation in App. C to confirm that this noise is indeed dependent across dimensions.

## 5.1 ELBO Results

We proceed to analyse the log-likelihood performance of relevant models on three image datasets: static Binary MNIST [38], Omniglot [24] as adapted by Burda et al. [8], and Fashion MNIST (fM-NIST) [44]. All models are trained with 32 latent variables using the same convolutional architecture as in Alemi et al. [3] except with ReLU activations. We trained using Adam optimization for 200 epochs, with an initial learning rate of 0.0003 decaying linearly to 0 over the last 100 epochs.

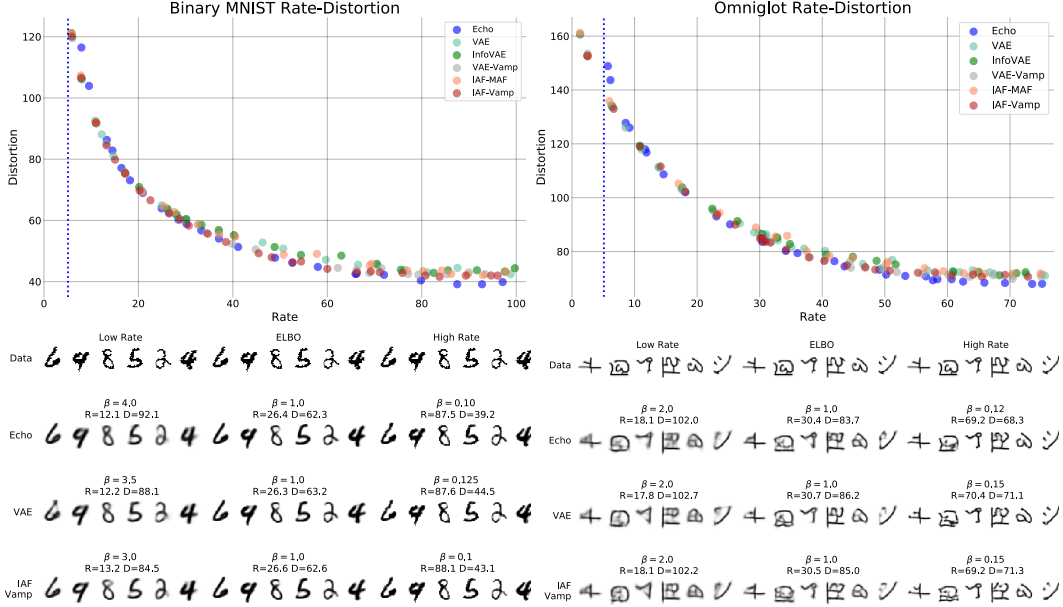

Figure 2: Binary MNIST R-D and Visualization    Figure 3: Omniglot R-D and Visualization

Table 1 shows negative test ELBO values, with the rate column reported as the appropriate upper bound for comparison methods. Results are averaged from ten runs of each model after removing the highest and lowest outliers. We compare Echo against diagonal Gaussian noise and IAF encoders, each with four marginal approximations: a Gaussian prior with and without the MMD penalty (e.g. *IAF-Prior*, *IAF+MMD*), MAF [31], and VampPrior [40]. Note that *VAE* is still used to denote the Gaussian encoder when paired with a different marginal (e.g. *VAE-Vamp*).

We find that the Echo noise autoencoder obtains improved likelihood bounds on Binary MNIST and Omniglot, with competitive results on fMNIST. We emphasize that Echo achieves this performance with significantly fewer parameters than comparison methods. IAF and MAF each require training an additional autoregressive model with size similar to the original network, while the VampPrior uses 750 learned pseudoinputs of the same dimension as the data. Although Echo involves special computation to construct the noise for each training example, it has the same number of parameters as a standard VAE and runs in approximately the same wall clock time.

We observe only minor differences based on the choice of encoding mechanism, which is somewhat surprising given the additional expressivity of the IAF transformation. The benefit of the flow transformations may be more readily observed on more difficult datasets or with more advanced architecture tuning [22].

We do find that a more complex marginal approximation can help performance. Although we see minimal gains from the MMD penalty and MAF marginal, the VampPrior bridges much of the performance gap with Echo noise. Recall that a learned prior can help ensure a tight rate bound while providing flexibility to learn a more complex marginal (in this case, a mixture model). However, the relative contribution of these effects is difficult to decouple. Echo instead provides both an exact rate and an adaptive prior by directly linking the choice of encoder and marginal.

## 5.2   Rate Distortion Curves

Moving beyond the special case of $\beta = 1$, rate-distortion theory provides the practitioner with an entire space of compression-relevance tradeoffs corresponding to constraints on the rate. We plot R-D curves for Binary MNIST in Fig. 2, Omniglot in Fig. 3, and Fashion MNIST in App. D.1. We also show model reconstructions at several points along the curve, with the output averaged over 10 encoding samples to observe how stochasticity in the latent space is translated through the decoder.

Table 2: Disentanglement Scores

Figure 4: Echo $\beta = 0, \gamma = 1$

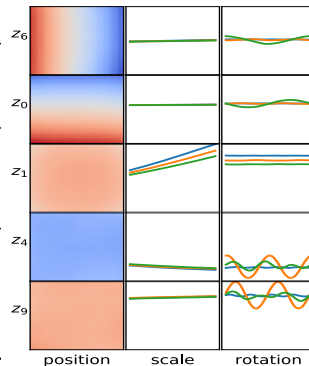

| | Independent Ground Truth | | | | Dependent Ground Truth | | | |
| | Factor | | MIG | | Factor | | MIG | |
| | Echo | VAE | Echo | VAE | Echo | VAE | Echo | VAE |
|---|---|---|---|---|---|---|---|---|
| $\beta = 1$ | **0.83** | 0.65 | 0.16 | 0.07 | **0.70** | 0.60 | 0.11 | 0.08 |
| $\beta = 4$ | 0.78 | 0.65 | 0.18 | 0.10 | 0.67 | 0.60 | 0.11 | 0.07 |
| $\beta = 8$ | 0.75 | 0.69 | 0.18 | 0.13 | 0.56 | 0.56 | 0.06 | 0.06 |
| $\gamma = 0$ | 0.83 | 0.65 | 0.16 | 0.07 | 0.70 | 0.60 | 0.10 | 0.08 |
| $\gamma = 20$ | 0.78 | 0.72 | **0.30** | 0.17 | 0.65 | 0.60 | **0.16** | 0.07 |
| $\gamma = 50$ | 0.79 | 0.73 | **0.30** | 0.18 | 0.58 | 0.53 | **0.16** | 0.07 |
| $\gamma = 100$ | 0.77 | 0.70 | 0.29 | 0.18 | 0.49 | 0.53 | 0.09 | 0.08 |

position      scale      rotation

These visualizations are organized to compare models with similar rates, which we emphasize may occur at different values of $\beta$ for different methods depending on the shape of their respective curves.

The Echo rate-distortion curve indeed exhibits several notable differences with comparison methods. We first note that Echo performance begins to drop off as we approach the lower limit on achievable rate, which is shown with a dashed vertical line and ensures that the rate calculation accurately reflects the noise for a finite number of samples (see App.B). In this regime, the sigmoids parameterizing $s_j(\mathbf{x})$ are saturated for much of training, and unused dimensions still count against the objective since we cannot achieve zero rate. We reiterate that this low rate limit may be adjusted by considering more terms in the infinite sum or decreasing the number of latent factors.

At low rates, our models maintain only high level features of the input image, and the blurred average reconstructions reflect that different samples can lead to semantically different generations. On both datasets, Echo gives qualitatively different output variation than Gaussian noise at low rate and similar distortion. Intermediate-rate models still reflect some of this sample diversity, particularly on the more difficult Omniglot dataset.

For very high capacity models, we observe that Echo slightly extends its gains on both datasets, with three to five nats lower distortion than comparison methods at the same rates. Intuitively, a more complex encoding marginal may be harder to match to a (learned) prior, loosening the upper bound on mutual information. The Echo approach can be particularly useful in this regime, as it avoids explicitly constructing the marginal while still providing exact rate regularization.

### 5.3 Disentangled Representations

Significant recent attention has been devoted to learning *disentangled* representations of data, which reflect the true generative factors of variation in the data [10, 27] and may be useful for downstream tasks [26, 42]. While prevailing definitions and metrics for disentanglement have recently been challenged [26], existing methods often rely on the inductive bias of independent ground truth factors, either via total correlation (TC) regularization [10, 20], or by using higher $\beta$ to more strongly penalize the KL divergence to an independent prior [9, 18]. Since Echo does not assume a factorized encoder or marginal, we investigate whether it can better preserve disentanglement when the ground truth factors are not independent.

To evaluate the quality of Echo noise representations, we compare against VAE models with diagonal Gaussian noise and priors, and consider the effects of increasing $\beta$ or adding independence regularization with parameter $\gamma$ [10, 20]:

$$\mathcal{L} = \mathbb{E}_q \log p_\theta(\mathbf{x}|\mathbf{z}) - \beta I_q(X;Z) - \gamma\, TC(Z)$$

TC regularization is implemented as in [20], where a discriminator is trained to distinguish samples from $q(\mathbf{z})$ and $\prod q(z_j)$. We keep $\beta = 1$ when modifying $\gamma$. Note that enforcing marginal independence will also limit the dependence in the noise learned by Echo, since $TC(Z|X)$ and $TC(Z)$ are linked as described in Sec. 2.1.

We calculate disentanglement scores on the dSprites dataset [28], where the ground truth factors of shape, scale, x-y position, and rotation are known and sampled independently across the dataset. To induce dependence in the ground truth factors, we downsample the dataset by partitioning each factor into 4 bins and randomly excluding pairwise combinations of bins with probability 0.15. This leads to a dataset of 15% of the original size, with a total correlation of 1.49 nats in the generative factors. We use both the implementation and experimental setup of Locatello et al. [26] and average scores over ten runs of each method.

Table 2 reports FactorVAE [20] and Mutual Information Gap [10] scores for both independent and dependent ground truth factors. We find that Echo provides superior disentanglement scores to VAEs across the board, although the relative improvement does not increase in the case of dependent latent factors. On the full dataset, independence regularization improves the MIG score for Echo and both scores for VAE, but may guide both models toward more entangled representations when this inductive bias does not match the ground truth. Finally, we note that increasing $\beta$ need not improve disentanglement for Echo noise, since we have relaxed assumptions of independence in both the encoder and marginal. Higher $\beta$ actually appear to hurt disentanglement scores on the dependent dataset for both methods.

In Figure 4, we visualize an Echo model that has successfully learned to disentangle position and scale, but not rotation, on the full dSprites dataset. Each row represents a single latent dimension, and each column shows mean $f(\mathbf{x})$ values as a function of the respective ground truth factors. Note that the first column shows a heatmap in the x-y plane, while the orange, blue, and green lines indicate ellipse, square, and heart, respectively (see [10]). In general, we observed that Echo models achieved their highest MIG scores on position, scale, and shape, with rotation often entangled across two or more dimensions.

## 6 Conclusion

VAEs can be interpreted as performing a rate-distortion optimization, but may be handicapped by their weak compression mechanism, independent Gaussian marginal assumptions, and upper bound on rate. We introduced a new type of channel, Echo noise, that provides a more flexible, data-driven approach to constructing noise and admits an exact expression for mutual information. Our results demonstrate that using Echo noise in autoencoders can lead to better bounds on log-likelihood, favorable trade-offs between compression and reconstruction, and more disentangled representations.

The Echo channel can be substituted for Gaussian noise in most scenarios where VAEs are used, with similar runtime and the same number of parameters. Echo should also translate to other rate-distortion problems via the choice of distortion measure, including supervised learning with the traditional Information Bottleneck method [2, 39] and invariant representation learning as in [30]. Exploring further settings where mutual information provides meaningful regularization for neural network representations remains an exciting avenue for future work.

## Footnotes

[1]Our approach is also easily adapted to multiplicative noise, such as in Achille and Soatto [1].

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
