[Supplementary Material · echo_supp.pdf]

# A  Rate-Distortion Theory

Given a source $X \sim q(\mathbf{x})$ and a distortion function $d : \mathcal{X} \times \mathcal{Z} \mapsto \mathbb{R}^+$ over samples and their codes $Z$, the rate-distortion function is defined as an optimization over conditional distributions $q(\mathbf{z}|\mathbf{x})$:

$$R(D) = \min_{q(\mathbf{z}|\mathbf{x})} I_q(X; Z) \quad \text{subj } \mathbb{E}_{q(\mathbf{x})q(\mathbf{z}|\mathbf{x})} d(\mathbf{x}, \mathbf{z}) \leq D \tag{9}$$

It is common to optimize an unconstrained problem by introducing a Lagrange multiplier $\beta^{-1}$ which, at optimality, reflects the tradeoff between compression and fidelity as the slope of the rate-distortion function at $D$, i.e. $\beta^{-1} = -\frac{\partial R}{\partial D}$:[2]

$$\mathcal{L} = \max_{\beta} \min_{q(\mathbf{z}|\mathbf{x})} I_q(X; Z) + \beta^{-1} \left( \mathbb{E}_{q(\mathbf{x})q(\mathbf{z}|\mathbf{x})} d(\mathbf{x}, \mathbf{z}) - D \right)$$

Eq. 8 suggests the cross entropy reconstruction loss as a distortion measure, so that $d(\mathbf{x}, \mathbf{z}) = -\log p_\theta(\mathbf{x}|\mathbf{z})$. We can then observe the equivalence between the rate-distortion optimization and our problem definition, as only the tradeoff between rate and distortion affects the characterization of solutions.

It is also interesting to note the self-consistent equations which solve the variational problem above (see, e.g. Tishby et al. [39])

$$q(\mathbf{z}|\mathbf{x}) = \frac{q(\mathbf{z})}{Z(\mathbf{x}, \beta)} \exp\left( -\beta^{-1} d(\mathbf{x}, \mathbf{z}) \right)$$

$$q(\mathbf{z}) = \int q(\mathbf{z}|\mathbf{x}) q(\mathbf{x}) d\mathbf{x}$$

Notice that, regardless of the choice of distortion measure, our Echo noise channel enforces the second equation *throughout* optimization by using the encoding marginal as the 'optimal prior.' For our choice of distortion, the solution simplifies as:

$$q(\mathbf{z}|\mathbf{x}) = \frac{q(\mathbf{z}) p(\mathbf{x}|\mathbf{z})^{1/\beta}}{Z(\mathbf{x}, \beta)} \tag{10}$$

This provides an interesting comparison with the generative modeling approach. While the Evidence Lower Bound objective can be interpreted as performing posterior inference with prior $p(\mathbf{z})$ in the numerator, we see that the information theoretic perspective prescribes using the exact encoding marginal $q(\mathbf{z})$. Indeed, our version of the ELBO bounds in Eq. 7 bounds the likelihood under the generative model $p(\mathbf{x}) = \int q_\phi(\mathbf{z}) p_\theta(\mathbf{x}|\mathbf{z}) d\mathbf{x}$. The gap in this bound then becomes $D_{KL}[q_\phi(\mathbf{z}|\mathbf{x}) || \frac{q(\mathbf{z})p(\mathbf{x}|\mathbf{z})}{Z(\mathbf{x})}]$, encouraging the encoder to match the rate-distortion solution for $\beta = 1$.

# B  Implementation of Echo Noise Sampling

Numerically, Gaussian noise cannot be sampled exactly and is instead approximated to within machine precision. We discuss several unique implementation choices that allow us to generate similarly precise Echo noise samples. In particular, we must ensure that the infinite sum defining the noise in Eq.3 converges and is accurately approximated using a finite number of terms.

**Activation Functions:** We parameterize the encoding functions $f(\mathbf{x})$ and $S(\mathbf{x})$ using a neural network and can choose our activation functions to satisfy the convergence conditions of Lemma 2.3. We let the final layer of $f$ use an element-wise $\tanh(\cdot/16)$ to guarantee that the magnitude is bounded: $\forall \mathbf{x}, |f(\mathbf{x})| \leq 1$. We found it useful to expand the linear range of the $\tanh$ function for training stability, although differences were relatively minor and may vary by application. One could also consider clipping the range of a linear activation to enforce a desired magnitude $|f(\mathbf{x})| \leq M$.

For the experiments in this paper, $S(\mathbf{x})$ is diagonal, with functions $s_j(\mathbf{x})$ on the diagonal. We implement each $s_j(\mathbf{x})$ using a sigmoid activation, making the spectral radius $\rho(S(\mathbf{x})) = \max_j |s_j(\mathbf{x})| \leq 1$. However, this is not quite enough to ensure convergence, as $\forall \mathbf{x}, s_j(\mathbf{x}) = 1$ would lead to an infinite

amount of noise. We thus introduce a clipping factor on $s_j(\mathbf{x})$ to further limit the spectral radius and ensure accurate sampling in this high noise, low rate regime.

**Sampling Precision:** When can our infinite sum be truncated without sacrificing numerical precision? We consider the sum of the remainder terms after truncating at $\ell = d_{\max}$ using geometric series identities. For $|f(\mathbf{x})| \leq M$ and $\rho(S(\mathbf{x})) \leq r$, we know that the sum of the infinite series will be less than $\frac{M}{1-r}$. The first $d_{\max}$ terms will have a sum given by $M\left(\frac{1-r^{d_{\max}}}{1-r}\right)$, so the remainder will be less than $M\left(\frac{r^{d_{\max}}}{1-r}\right)$. For a given choice of $d_{\max}$, we can numerically solve for $r$ such that the sum of truncated terms falls within machine precision $M\left(\frac{r^{d_{\max}}}{1-r}\right) \leq 2^{-23}$. For example, with $M = 1$ and $d_{\max} = 99$, we obtain $r = 0.8359$. We therefore scale our element-wise sigmoid to $s_j(\mathbf{x}) = r\sigma(\cdot)$ for calculating both the noise and the rate.

**Low Rate Limit:** This clipping factor limits the magnitude of noise we can add in practice, and thus defines a lower limit on the achievable rate in an Echo model. For diagonal $S(\mathbf{x})$, the mutual information can be bounded in terms of $r$, so that $I(X; Z) = -\sum_{j=1}^{d_z} \mathbb{E}_q \log |s_j(\mathbf{x})| \geq -d_z \log r$. Note that $r$ is increasing in $d_{\max}$, since the first term in the remainder decreases exponentially with the number of terms. Each included term can then have higher magnitude, leading to lower achievable rates. Thus, this limit can be tuned to achieve strict compression by increasing $d_{\max}$ or simply using fewer latent factors $d_z$.

**Batch Optimization:** Another consideration in choosing $d_{\max}$ is that we train using mini-batches of size $B$ for stochastic gradient descent. For a given training example, we can use the other iid samples in a batch to construct Echo noise, thereby avoiding additional forward passes to evaluate $f$ and $S$. There is also a choice of whether to sample with or without replacement, although these will be equivalent in the large batch limit. In experiments we saw little difference between these strategies, and proceed to sample without replacement to mirror the treatment of training examples. We let $d_{\max} = B - 1$ to set the rate limit as low as possible for this sampling scheme.

## C  Total Correlation for Echo Noise

To briefly demonstrate that Echo noise is dependent across latent dimensions, we can estimate the total correlation of noise samples in Table 3 using the second-order covariance approximation $TC(\epsilon) = -\log|\Sigma_{\mathrm{diag}_\epsilon^{-1}} \Sigma_\epsilon|$. This is clearly zero for diagonal Gaussian noise, and provides a sufficient condition to show that the Echo noise is not independent.

Table 3: TC by Dataset

|            | Binary MNIST | Omniglot | Fashion MNIST |
|------------|--------------|----------|---------------|
| $TC(\epsilon)$ | 7.3      | 18.8     | 30.2          |

For the Echo models considered in this work, we can also derive an interesting equivalence between the conditional and overall total correlation. Observe that the expression for mutual information in Eq. 4 decomposes for diagonal $S(\mathbf{x})$:

$$I_q(X; Z) = -\mathbb{E}_{q(\mathbf{x})} \log |\det S(X)|$$

$$= -\mathbb{E}_{q(\mathbf{x})} \sum_{j=1}^{d_z} \log s_j(X)$$

This additivity across dimensions implies that $I_q(X; Z) = \sum_{j=1}^{d_z} I_q(X; Z_j)$. Before proceeding, we first recall the definitions of total correlation and conditional total correlation [43], which measure the divergence from independence of the marginal and conditional, respectively:

$$TC(Z) = D_{KL}[q_\phi(\mathbf{z}) || \prod_{j=1}^{d_z} q_\phi(z_j)]$$

$$TC(Z|X) = D_{KL}[q_\phi(\mathbf{z}|\mathbf{x}) || \prod_{j=1}^{d_z} q_\phi(z_j|\mathbf{x})]$$

Now consider the quantity $D_{KL}[q_\phi(\mathbf{z}|\mathbf{x})||\prod_{j=1}^{d_z} q_\phi(z_j)]$. We can decompose this in two different ways, first by projecting onto the joint marginal:

$$D_{KL}[q_\phi(\mathbf{z}|\mathbf{x})||\prod_{j=1}^{d_z} q_\phi(z_j)] = \mathbb{E}_q \log \frac{q_\phi(\mathbf{z}|\mathbf{x})}{\prod_{j=1}^{d_z} q_\phi(z_j)}$$

$$= \mathbb{E}_q \log \frac{q_\phi(\mathbf{z}|\mathbf{x})}{\prod_{j=1}^{d_z} q_\phi(z_j)} \frac{q_\phi(\mathbf{z})}{q_\phi(\mathbf{z})}$$

$$= I_q(X;Z) + TC(Z)$$

We can also decompose using the factorized conditional:

$$D_{KL}[q_\phi(\mathbf{z}|\mathbf{x})||\prod_{j=1}^{d_z} q_\phi(z_j)] = \mathbb{E}_q \log \frac{q_\phi(\mathbf{z}|\mathbf{x})}{\prod_{j=1}^{d_z} q_\phi(z_j)}$$

$$= \mathbb{E}_q \log \frac{q_\phi(\mathbf{z}|\mathbf{x})}{\prod_{j=1}^{d_z} q_\phi(z_j)} \frac{\prod_{j=1}^{d_z} q_\phi(z_j|\mathbf{x})}{\prod_{j=1}^{d_z} q_\phi(z_j|\mathbf{x})}$$

$$= \sum_{j=1}^{d_z} I_q(X;Z_j) + TC(Z|X)$$

The equality of $I(X;Z)$ and $\sum_{j=1}^{d_z} I_q(X;Z_j)$ implies equality for $TC(Z)$ and $TC(Z|X)$.

$$I_q(X;Z) + TC(Z) = \sum_{j=1}^{d_z} I_q(X;Z_j) + TC(Z|X)$$

$$\implies TC(Z) = TC(Z|X)$$

The effects of this relationship have not been widely studied, as $TC(Z|X) = 0$ for traditional VAE models. On the other hand, $TC(Z)$ is usually non-zero and has been minimized as a proxy for 'disentanglement' [20, 10]. We evaluate similar regularization for Echo in Sec. 5.3.

We have shown that parallel Echo channels are perfectly additive in that $\sum_j I_q(X;Z_j) - I_q(X;Z) = 0$. However, general channels could be sub- or super-additive, so that $TC(Z) < TC(Z|X)$, $TC(Z) = TC(Z|X)$, or $TC(Z) > TC(Z|X)$ (e.g. Sec. 4.2 of [17]). Extending Echo to non-diagonal $S(\mathbf{x})$ could allow us to explore the various relationships between $TC(Z)$ and $TC(Z|X)$ and more precisely characterize those which are useful for representation learning.

# D   Additional Results

## D.1   Fashion MNIST Rate-Distortion

We show a full rate-distortion curve for Fashion MNIST in Fig.5, along with reconstructions at various rates. Echo performance nearly matches that of comparison methods except at low rates.

Figure 5: FMNIST Rate-Distortion and Visualization

## D.2   Marginal Activations

We visualize dimension-wise marginal activations for Echo on Binary MNIST and Omniglot in Fig.6. We show $q(z_j)$ for thirteen dimensions in each method, including nine with highest rates, three with low rates, and one with minimal rate. For each, we combine activations from 2000 encoder samples on each test example and fit a KDE estimator with RBF bandwidth chosen according to the Scott criterion.

As discussed in Sec. 2, Echo avoids assumptions that the marginals are independent and Gaussian as in VAEs. However, we observe the individual Echo marginals $q(z_j)$ to be approximately Gaussian, with the Anderson-Darling test failing to reject the null hypothesis of Gaussianity for any dimension. Nevertheless, the joint marginal $q(\mathbf{z})$ may still be dependent (see App. C).

Individual dimensions are also are free to learn different means and variances without incurring a penalty in the objective, with factors generally keeping more mutual information with the data having less variance in the marginals. The highest mean dimension in the Omniglot plot corresponds to an 'unused' dimension that saturates the lower limit on achievable rate.

Figure 6: Marginal Activations by Dimension

## D.3   Echo $f(\mathbf{x})$ vs. $S(\mathbf{x})$

We can analyse the Echo mutual information at each data point by noting that the expression in Eq. 4 involves an expectation over $\mathbf{x}$. Since $H(Z)$ and $H(\mathcal{E})$ do not depend on $X$ in the proof of Thm. 2.2, we can evaluate $-\sum_j \log s_j(\mathbf{x})$ as a pointwise mutual information. We compare this quantity with the L2-norm of $f(\mathbf{x})$ as a proxy for signal to noise ratio. Test examples are sorted by conditional likelihood $p_\theta(\mathbf{x}|\mathbf{z})$ on the x-axis, and we see that Echo indeed has higher mutual information on examples where the generative model likelihood is high. Further analysis of these pointwise informations remains for future work.

Figure 7: Echo $f(\mathbf{x})$ vs. $S(\mathbf{x})$: Binary MNIST (left) and Omniglot (right)

# E  Details for Experiments

All models were trained using a similar convolutional architecture as used in [3], but with ReLU activations, unnormalized gradients, and fewer latent factors. We use Keras notation and list convolutional layers using the arguments (filters, kernel size, stride, padding). We show an example parametrization of Echo in the hidden layer.

- Conv2D(32, 5, 1, 'same')
- Conv2D(32, 5, 2, 'same')
- Conv2D(64, 5, 1, 'same')
- Conv2D(64, 5, 2, 'same')
- Conv2D(256, 7, 1, 'valid')
- `echo_input` = [Dense(32, `tanh`($\cdot/16$)), Dense(32, `tf.math.log_sigmoid`)]
- Lambda(`echo_sample`)(`echo_input`)
- Conv2DTranspose(64, 7, 1, 'valid')
- Conv2DTranspose(64, 5, 1, 'same')
- Conv2DTranspose(64, 5, 2, 'same')
- Conv2DTranspose(32, 5, 1, 'same')
- Conv2DTranspose(32, 5, 2, 'same')
- Conv2DTranspose(32, 4, 1, 'same')
- Conv2D(1, 4, 1, 'same', activation = 'sigmoid')

We trained using Adam optimization for 200 epochs, with a learning rate of 0.0003 decaying linearly to 0 over the last 100 epochs. All experiments were run using NVIDIA Tesla V100 GPUs.

MAF and IAF models were implemented using the Tensorflow Probability package [13]. Each uses four steps of mean-only autoregressive flow, with each step consisting of three layers of 640 units. For the VampPrior, we used 750 pseudoinputs on all datasets. For the *IAF-Vamp* experiments, note that the VampPrior is calculated with respect to the inputs $z_0$ of the IAF transformation to avoid expensive density evaluations on new samples. This is valid since the mean-only transformation has constant Jacobian, but makes this method closely resemble VAE-Vamp. All MMD penalties had a loss coefficient of 999, and were evaluated using a radial basis kernel with bandwidth $\sigma = 32/\sqrt{2}$ as in [45, 46].

For rate-distortion experiments, we evaluated $\beta = [.05, .075, .1, .125, .15, .2, .25, .3, .4, .5, .6, .7, .8, .9, 1, 1.5, 2, 3, 4, 6]$, with additional $\beta$ to fill in gaps in the curve as necessary.

For the disentanglement experiments in Sec. 5.3, we followed the architecture and hyperparameters in Locatello et al. [26]. We trained for 300,000 gradient steps on both the full dataset and the downsampled dataset with dependent factors. The visualization in Figure 4 was generated using code from Chen et al. [10].

Code implementing these experiments can be found at `https://github.com/brekelma/echo`.

## Footnotes

[2]Note, we have constrained the distortion here, instead of the rate as in the main text. We write the Lagrange multiplier as $\beta^{-1}$ to maintain a correspondence between the parameterizations of each problem.