[Reviews · NeurIPS 2019]

Reviewer 1



[Update after taking into account author feedback: thanks for preparing the author feedback and addressing the issues raised by the reviewers. Given the clarifications and additional results, I am raising my score to 7 and vote and argue for accepting the paper. ] The paper proposes a novel stochastic encoder-channel, Echo Noise, that allows for an analytically tractable expression of the latent marginal distribution without imposing strong assumptions about the latent-space distribution, which is in contrast to e.g. the commonly used independent Gaussian assumption. Importantly, the Echo Noise channel allows for a simple analytical expression of the mutual information (between input and encoded/latent variable) that can be used as a regularizer in a mini-batch based gradient optimisation scheme. Applying Echo Noise in the context of auto-encoding, leads to several theoretically appealing properties, most importantly it can be shown that Echo Noise automatically ensures that the latent-space prior matches the marginal, which is the “optimal” prior and allows for a simple and tractable implementation of the (exact) rate-distortion objective instead of a “looser” general-prior ELBO. Related literature is discussed in a concise and well-written manner, and some results on simple image-auto-encoding benchmarks are reported and compared against a number of state-of-the-art methods. In the recent literature the connection between lossy compression (i.e. rate-distortion) and representation learning has been pointed out several times, in particular w.r.t. interpreting the evidence lower bound as a degenerate, or suboptimal, version of the rate-distortion objective that can be improved upon by using the “optimal” prior. Unfortunately, common assumptions for the functional form of the prior and/or latent-space distribution for reasons of tractability have often prevented a straightforward implementation of the rate-distortion objective (as a tighter ELBO). This paper presents an interesting approach to tackle this problem, with an iterative stochastic encoder that is simple to implement and does not seem to add substantial computational overhead. This is an important step towards better objective functions and a more solid theoretical basis for representation learning, not only in the self-supervised but potentially also in the supervised setting (-> Info Bottleneck). I personally find the topic very timely, and the proposed approach both novel and original. Echo Noise is nicely introduced and its appealing theoretical properties are analyzed and presented in a solid fashion (given access to the appendix). A big plus is also that the authors analyzed convergence behavior and bounds for approximating the infinite-series construction of Echo Noise via a finite-iteration procedure. Unfortunately I found the experimental section a bit underwhelming and thin. While I appreciate the comparison with many other state-of-the-art methods on some standard auto-encoding benchmarks, I would have been very excited to see some thorough analysis of the learned representations, e.g. some exploration of the latent space distribution, which should now be of a much less restricted form than many competitor methods. It would have been great to see, e.g. structure and topology of the latent space (under different rate-limits), some inter- or extrapolations, disentanglement, or some experiments in the supervised setting, or at least some secondary tasks to assess the quality of the learned representations (after all the paper’s aim is representation learning). I personally do not find comparing neg. log likelihood scores very insightful when talking about representation learning (the log likelihood/reconstruction error says a lot more about the generative model than the quality of the learned representation) - however, I do recognize that this is commonly used as a proxy in many publications. But particularly in the low-rate regime (where representation learning is particularly interesting), the structure of the latent space and potential transferability of learned representations becomes much more interesting than log-likelihood scores or blurry generated samples (perhaps more emphasis on the multimodal nature of the generated samples in the low-rate regime would help a bit). I think that this paper had all the right ingredients in place to show some exciting results, but ends up with a somewhat underwhelming experimental section. As much as I would like to give the first four sections of the paper a very high score, the current experimental section brings my overall score down to 6, but I’m happy to be convinced otherwise during the rebuttal or by the other reviewers. Originality: high for Echo Noise construction, low for experimental section Quality: Very well written paper, high-quality literature overview and discussion of related methods, some nice (theor.) results, connections and analysis in the appendix. Clarity: Writing is clear, mathematics support the main arguments and are certainly not used to obfuscate quality or novelty, unfortunately the appendix is missing in the upload - since I had already seen the preprint of the paper before reviewing (which is admittedly not ideal), I referred to the appendix in the preprint version. Significance: Potential for high impact, mostly due to alleviating very limiting requirements on the latent space distribution/prior and by taking the rate-distortion viewpoint on repres. learning seriously. Unfortunately the experiments do not show too much of the promising advantages (at least not very directly), which leaves the current significance a bit more questionable and perhaps at a medium level.

Reviewer 2



This is a really wonderful paper, in moving away from perhaps naive uses of Gaussian distributions in VAEs to the so-called echo distribution that has both theoretical and practical benefits. The paper also solidifies the rate-distortion view of VAE and yields distributions with closed-form mutual information expressions. The topical area of VAEs is certainly important and so this is a great contribution for the community. The analytical idea for constructing echo noise is also quite clever. The paper is generally well-written and the experiments are also well-performed. As far as I can tell, this work is quite novel.

Reviewer 3



Personally, I'm not an expert in the topic covered in this paper, so my review should probably be downweighed. Having said that, the theoretical contributions seem weak. The conditions in Lemma 2.3 are not discussed and so the implementation of the echo noise mechanism in (3) may be unclear in practice. Furthermore, there is also no discussion on why we can restrict S(x) to be diagonal. One of the claims the authors make is that prior assumptions on Gaussianity allow for analytical tractability. By assuming that S(x) is diagonal, aren't the authors also making assumptions for analytical tractability to ensure that the mutual information can be computed in closed-form? The relation to rate-distortion theory in classical information theory is also very weak. In particular, where is the distortion measure? The claim that Echo has "approximately the same wall time" as VAEs should be substantiated with evidence. Currently, this claim is weak. In general, this paper is not very convincing.

[Author Response · NeurIPS 2019]

We thank the reviewers for their time and comments. Mutual information has emerged has an indispensable tool in representation learning and we share the reviewers' enthusiasm to explore further implications of our method, the first to provide an analytic expression for the compression rate for arbitrary input distributions. Some reviews express concern that we do not elucidate the breadth of potential applications for our approach, so we briefly discuss some of these ideas before addressing more detailed response points.

The supervised Information Bottleneck is a natural first extension, and we can indeed reference our concurrent work using Echo in the context of fairness and invariant representation learning. For example, we train using the IB functional to classify digits from an augmented MNIST dataset with rotation at angles $\{0, \pm 22.5, \pm 45\}$. We then train a classifier to predict rotation (a nuisance) from our representation, shown in Table 1. We find that Echo learns close to a task-minimal, invariant representation (since the classifier is simply random chance), and improves upon several baselines [2, 5]. We show similar results on the 3d Chairs dataset, with style as the label and yaw angle as the desired invariance.

Reviewer 1 references disentanglement as another desirable property in representation learning, although recent work has questioned current approaches and definitions of this term [4]. In the setting of [4], we calculate ELBOs and disentanglement scores ([3]) for Echo and several comparisons in Table 2. With no modifications of the objective and diagonal $S(x)$, Echo obtains competitive disentanglement scores with superior reconstruction. We plan to strengthen these results in the final version and extend our evaluation to the case when the ground truth factors of variation are dependent. In this scenario, Echo may have more flexibility to preserve disentanglement than independence-based methods [1, 3] given its lack of such assumptions.

While extending Echo to non-diagonal (e.g. triangular) $S(x)$ is an ongoing next step, we chose to first analyze the diagonal case where the architecture is directly comparable to a VAE (with the same structure and number of parameters). Further, the structure of $S(x)$ does not affect our ability to derive an analytic mutual information for arbitrary data, whereas common VAE assumptions only yield an exact rate in the unrealistic setting of Gaussian inputs. As we note in Sec. 2.1, Echo noise may still be dependent across dimensions with diagonal $S(x)$, and maintains tractability without prescribing a Gaussian form for the encoder or marginal.

With respect to the conditions of Lemma 2.3, bounding the activations $|f(x)| < M$ and $|s_j(x)| < r$ leads to a straightforward argument for convergence of the infinite sum defining Echo noise. If we truncate after $d$ samples, we can bound the sum of remainder terms to be within machine precision by solving for $r$ s.t. $M r^d/(1 - r) < 2^{-23}$. We will clarify this reasoning in the main text, with detailed discussion of our implementation in the Appendix.

To address other minor comments, we found that Echo ran in just under 110% of the wall clock time for VAE on an NVIDIA Tesla V100 GPU, with a consistent ratio across datasets. While our distortion measure is referenced in line 163, we will emphasize this choice in Sec. 3.1. Our Appendix develops more detailed connections with classical rate-distortion and recent related works, which we hope will form a rich foundation for future developments.

Table 1: Invariant Classification Results
Label (higher is better) and Nuisance (lower is better)
Random chance : 0.20 for MNIST, 0.25 for Chairs

| Method | MNIST-Rotated | | Chairs | |
|---|---|---|---|---|
| | Label | Nuisance | Label | Nuisance |
| Echo IB | **0.98** | **0.20** | **0.84** | **0.25** |
| UAI [2] | 0.98 | 0.34 | 0.74 | 0.34 |
| VFAE [5] | 0.95 | 0.38 | 0.72 | 0.37 |

Table 2: FactorVAE Disentanglement Scores: dSprites [3]

| | ELBO | Disentanglement |
|---|---|---|
| Echo $\beta = 1$ | **41.8** | .66 |
| VAE $\beta = 1$ | 46.7 | .61 |
| Factor-VAE $\gamma = 10$ [3] | 62.0 | .72 |
| Factor-VAE $\gamma = 50$ [3] | 74.2 | **.73** |

# References

[1] Tian Qi Chen, Xuechen Li, Roger B Grosse, and David K Duvenaud. Isolating sources of disentanglement in variational autoencoders. In *Advances in Neural Information Processing Systems*, pages 2610–2620, 2018.

[2] Ayush Jaiswal, Rex Yue Wu, Wael Abd-Almageed, and Prem Natarajan. Unsupervised adversarial invariance. In *Advances in Neural Information Processing Systems*, pages 5092–5102, 2018.

[3] Hyunjik Kim and Andriy Mnih. Disentangling by factorising. *arXiv preprint arXiv:1802.05983*, 2018.

[4] Francesco Locatello, Stefan Bauer, Mario Lucic, Sylvain Gelly, Bernhard Schölkopf, and Olivier Bachem. Challenging common assumptions in the unsupervised learning of disentangled representations. *arXiv preprint arXiv:1811.12359*, 2018.

[5] Christos Louizos, Kevin Swersky, Yujia Li, Max Welling, and Richard Zemel. The variational fair autoencoder. *arXiv preprint arXiv:1511.00830*, 2015.


[Meta-Review · NeurIPS 2019]

This submission was thoroughly reviewed and discussed among the reviewers. The conclusion is that the paper contains some interesting ideas, but it was also judged weak technically by one of the reviewer. The latter makes valid points. The manifest interest in this paper and its relevance for NeurIPS is currently leaving the door open for acceptance, but I suggest an additional review to confirm that.